# A socio-ecological System Dynamics model of antimicrobial use and resistance

**Carys J. Redman-White**[1,2]*, **Birgit Kopainsky**[3], **Adrian Muwonge**[2], **Andrew R. Peters**[4], **Dominic Moran**[1]

**1** Global Agriculture and Food Systems, Royal (Dick) School of Veterinary Studies, University of Edinburgh, Edinburgh, United Kingdom, **2** Digital One Health Lab, The Roslin Institute, University of Edinburgh, Edinburgh, United Kingdom, **3** System Dynamics Group, Department of Geography, University of Bergen, Bergen, Norway, **4** Royal (Dick) School of Veterinary Studies, University of Edinburgh, Edinburgh, United Kingdom

\* c.j.redman-white@sms.ed.ac.uk

## Abstract

The concept of resource extraction in the context of antimicrobial resistance (AMR) is rarely explored. In this framework, antimicrobial susceptibility is viewed as a finite resource that is depleted through the use of antimicrobials —thus, AMR represents the exhaustion of this resource. In this conceptual study, we examine the system dynamics of AMR using causal loop diagrams to define both the structure and behaviour of two variants of the system. We then evaluate the robustness of the System Dynamics models through sensitivity testing. The first model is inspired by a classic "Limits to Growth" structure, in which antimicrobial use practices depend on recent observations of treatment success or failure. The second differs by including AMR surveillance informing antimicrobial prescribing instead of anecdotal experience. The models consider one "bug-drug-mutation-context" combination at a time, but can be applied to different microbes, antimicrobials, and host populations in human or veterinary contexts. Multiparameter sensitivity analyses of relative fitness and timescale parameters were carried out on both model variants. Several key differences in model behaviour over time were observed between the socio-ecological Limits to Growth structure and the modified version in which human judgment, with its associated time lag, is bypassed. The models help explore the effects of human behaviour and associated time lags on patterns of population-level antimicrobial susceptibility over time, applying an established modelling technique in a novel context to generate new hypotheses – notably that plateaux in AMR observed in the field could be in part due to human behaviour rather than purely evolutionary forces. The framing of antimicrobial susceptibility as a variably renewable natural resource is valuable not only as an avenue for alternative modelling approaches, but also as a more collaborative framing for public and policy engagement to promote sustainable management of this common pool resource.

**Data availability statement:** All relevant data are within the manuscript and its Supporting Information files. Model files are also available at https://doi.org/10.6084/m9.figshare.31792000.

**Funding:** CRW acknowledges support for a studentship part-funded by Zoetis and the UKRI Biotechnology and Biological Sciences Research Council (BBSRC) under grant number BB/T00875X/1. DM also acknowledges support under UKRI awards UKRI (BB/Z515644/1, BB/T004436/1 and BB/T004452/1, the latter of which is [co] funded by the UK Department of Health and Social Care as part of the Global AMR Innovation Fund (GAMRIF). This is a UK aid programme that supports early-stage innovative research in underfunded areas of antimicrobial resistance (AMR) research and development for the benefit of those in low and middle-income countries (LMICs), who bear the greatest burden of AMR. The views expressed in this publication are those of the author(s) and not necessarily those of the UK Department of Health and Social Care. AM is funded by the University of Edinburgh Chancellor's Fellowship and BBSRC core funding for the Roslin Institute. For the purpose of open access, the authors have applied a Creative Commons Attribution (CC BY) licence to any Author Accepted Manuscript version arising from this submission. The funders had no role in study design, data collection and analysis, decision to publish, or preparation of the manuscript.

**Competing interests:** The authors have declared that no competing interests exist.

## Introduction

Antimicrobial use (AMU) is vital to modern medicine and has facilitated huge increases in agricultural productivity over the last century, but exposure of human, animal and environmental microbiota to antimicrobial drugs inevitably creates an evolutionary selection pressure for the development and spread of antimicrobial resistance (AMR). The global health and economic consequences of this are well documented [1,2].

AMR can be mitigated but not eliminated by careful antimicrobial stewardship, and the reduction of AMR gene prevalence in a microbial population when an antimicrobial drug is withdrawn depends on the fitness cost of resistance, if any. Whilst these general principles apply across One Health (clinical, veterinary and environmental) contexts, there is great variation in epidemiology, microbiology, AMU practices and other key AMR determinants between microbes, antimicrobials, mutations, and One Health settings. This heterogeneity presents a major challenge in understanding the dynamics of AMR, in particular the quantitative relationship between AMU and AMR.

### Antimicrobial susceptibility as a natural resource

Discussion around AMU and antimicrobial efficacy focuses on resistance, the rhetoric frequently framing AMR as an enemy to be eliminated. But as a naturally occurring phenomenon pre-dating the use of antimicrobial drugs by humans [3], framing AMR as depletion of antimicrobial susceptibility, a natural resource that can be collaboratively exploited by use of antimicrobials, is a potentially more constructive narrative for motivating policy and public engagement. In this framing, susceptibility to a given antimicrobial is depleted as a result of selection pressure when the microbial population is exposed to the drug in question. Depending on selection pressures, the fitness landscape, and particularly the fitness cost (if any) of the mutation conferring resistance in the absence of the antimicrobial, susceptibility may behave as a renewable resource. This conceptual approach has received limited attention, with a small number of economic resource models of AMR published over two decades ago, approaching susceptibility as either a renewable or non-renewable resource [4,5].

Conceptual models can be substantiated using different modelling techniques to investigate AMU-AMR dynamics, generating and testing hypotheses to explain phenomena such as plateauing AMR [6,7]. In addition to focusing on evolutionary epidemiology within and between hosts [6], models are useful for exploring policy interventions and their potential impacts at the population scale [8]. This paper further explores the resource extraction framing using System Dynamics (SD) modelling to illustrate the interplay of key variables determining the renewable resource properties of susceptibility. Causal loop diagrams (CLDs) are used here to explore model structures, which form the basis of quantitative SD models. These are investigated using sensitivity testing to assess the behaviour and robustness of the model across a wide parameter space.

The next section introduces the SD approach and its use in AMR modelling, as well as examining previous resource extraction modelling approaches to AMR. A

methods section describes how this model and its structural variants were developed and the sensitivity analysis carried out to investigate their behaviour. This includes a version in which AMU is determined by anecdotal results of recent treatments and one in which it is determined by current surveillance of AMR. In a results section, the model structures and equations are presented along with the results of the sensitivity analysis. Finally, we discuss the implications of our findings and explore potential next steps.

## System archetypes

The systems thinking approach has been applied to a wide range of problems since its development in the late 20th century, with recurring problems in systems identified at a structural level. These problematic structures, or system archetypes, are well characterised and modelled [9]. One of the best known, the "Limits to Growth" (LTG) archetype, which lends its name to the title of the seminal 1972 report by the Club of Rome [10], describes how exponential growth can occur until it is limited by an external constraint. Common applications include harvesting of renewable and non-renewable resources such as fisheries and fossil fuel reserves, respectively. Considering antimicrobial susceptibility as a natural resource "harvested" by use of antimicrobial drugs, this may permit an application of this well-characterised modelling approach to the phenomenon of AMR. In this context, microbial susceptibility to a specific antimicrobial drug would be a partially renewable resource, with development of novel antimicrobial drugs analogous to accessing new "stocks" of the resource.

When represented using a CLD, LTG is a simple structure consisting of two feedback loops (Fig 1). In the case of a fishery, increased fishing (efforts) would lead to greater harvests and profit (performance), encouraging more fishing: a positive feedback (reinforcing) loop. However, this would deplete the fish stocks (limiting action/resource), which are replenished dependent on the species reproductive rate (constraint). Depleted stocks reduce harvests: a negative feedback (balancing) loop. Depending on the parameters, including time lags, this system can lead to "boom and bust" oscillations in the resource or may stabilise at an equilibrium [9].

## Mathematical modelling of AMR

Mathematical modelling of AMR at the population level has been approached in a variety of ways. Studies have included deterministic and stochastic models using compartmental and, to a lesser extent, individual based models, with an overall

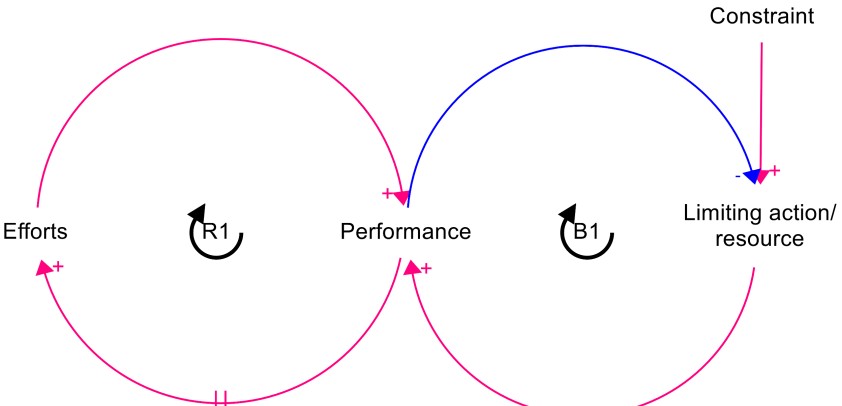

**Fig 1. Causal loop diagram of the "Limits to Growth" archetypal structure.** Arrows with a + cause variables to change in the same direction, while arrows with a – cause variables to change in the opposite direction. Feedback loops are labelled R for reinforcing (positive feedback) or B for balancing (negative feedback).

tendency to focus primarily on human health [8]. Compartmental models can be used to represent either populations of hosts colonised by resistant, susceptible and intermediate microbes or the microbe populations themselves across host species, for example to investigate the potential interactions between AMU and AMR in veterinary and human medicine contexts [11].

Several authors have combined compartmental models with economic approaches, modelling AMR as a natural resource. Two related models have emphasised different aspects of the economic and epidemiological challenge [4,5]. They combine susceptible-infected-susceptible (SIS) compartmental models [12] with economic resource extraction modelling, treating susceptibility as either a renewable or a non-renewable resource. The first model considers antimicrobial susceptibility as non-renewable and investigates the optimal usage of antimicrobials from a choice of two drugs with different costs, incorporating other economic parameters such as marginal benefit of successful treatments and discounting of the value of future successful treatments [4]. The authors allude to but do not explore the potential for cyclicity in cases in which susceptibility is renewable. The authors aim to identify the optimal AMU policy rather than investigate the behaviour of the system over time. The second model, based on the first, introduces fitness costs to resistance, modelling susceptibility as a renewable resource to investigate the impacts of strategies involving aggressive use of antimicrobials in comparison to focusing on managing susceptibility prevalence [5]. This paper also focuses on optimisation and steady-state outcomes, and while it does essentially model a LTG system, it does not incorporate any delays or investigate potential for oscillations in susceptibility over time or impacts of human "bounded rationality" on outcomes.

**1.3.1. System Dynamics modelling.** Whilst a variety of LTG systems have been quantitatively modelled using System Dynamics (SD), this framing has not been applied to AMR. Limited SD modelling of AMR has investigated the evolutionary dynamics of AMR in *Streptococcus pneumoniae* in humans in response to penicillin prescriptions [13]. This paper maps interlinkages between social, economic and policy influences on AMU and resistance, and discusses microbiological components, such as fitness costs of AMR varying between microbial strains. The authors present a quantitative model of AMU and resistance in *S. pneumoniae*, with the bacterial population modelled as densities of "susceptible", "intermediately resistant" and "highly resistant" subpopulations, independent of colonisation of specific hosts. This SD model specifically focuses on the population dynamics of AMR in response to the selection and fitness landscape, with AMU an exogenous variable independent of AMR. Consequently, factors influencing AMU are not investigated, leaving scope for development of models to investigate the behavioural aspects of AMU in the context of AMR. In an LTG model of AMR, specifically, the use of antimicrobials is influenced by observed successful treatment (Fig 2A).

## Methods

### Development of model structure

CLDs are visual representation of systems that allow us to explain how and why systems behave the way they do over time, by visualising how variables influence each other through reinforcing (positive) or balancing (negative) feedback loops. A CLD of AMR as a LTG system was developed by mapping the archetypal variables (Fig 1) to their corresponding variables for this application, with a single "bug-drug-mutation-context" combination considered. In this case, "efforts" refer to AMU, "performance" to cases successfully treated with this antimicrobial, the "limiting resource/action" to antimicrobial susceptibility, and the "constraint" to the rate at which microbes revert to the susceptible wild-type. A modification was made to the structure to reflect the fact that susceptibility is depleted as a result of selection pressure produced by AMU regardless of whether the case is successfully treated (Fig 2A). A variation on the model was created with surveillance bypassing the reinforcing loop (Fig 2B) and both versions of the CLD were used as the bases for quantitative SD models, in which the susceptible and resistant fractions of the microbial population are represented as stocks, as are the proportion of the host population currently being treated with the antimicrobial (Fig 3). The chosen model structure elides socio-behavioural drivers of AMU, such as costs, access, and prescribing norms. Whilst these may represent important

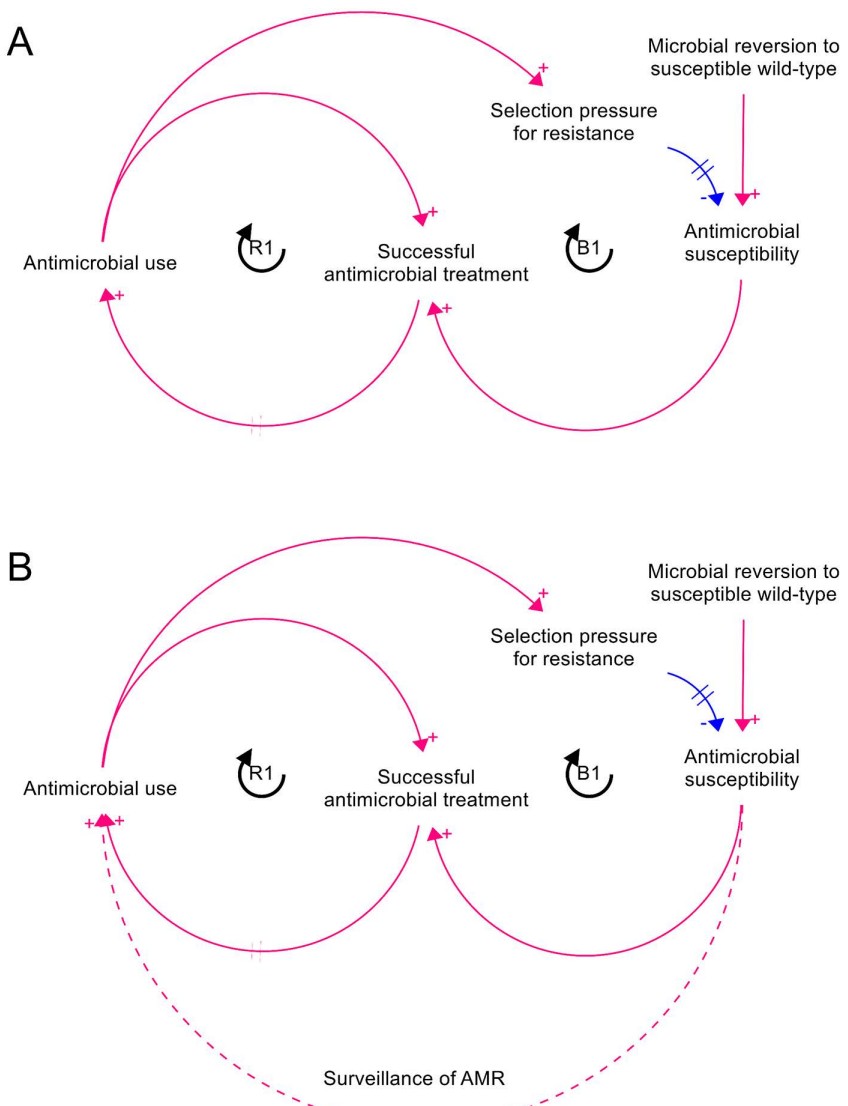

**Fig 2. CLD representing AMR as a Limits to Growth archetype.** Archteype shown a) in its basic form and b) with surveillance providing an information connector between susceptibility and AMU.

determinants of AMU in clinical and agricultural settings, the decision was made in this case to focus on the LTG feedback loop structure rather than to attempt to achieve full behavioural realism. All CLD and SD modelling was carried out using Stella Architect software [14].

**2.1.1. Defining renewability.** In this conceptualisation, the defining feature of susceptibility as a renewable resource is that resistance comes with a fitness cost, such that susceptibility "replenishes" when microbes are no longer exposed to the modelled antimicrobial. The degree of renewability is quantified in the model presented here as the replacement or reversion rate of the resistant form in the absence of the antimicrobial, which as it approaches zero represents a non-renewable stock of susceptibility. In evolutionary terms, in order to be considered renewable, the fitness of the resistant form in the absence of the antimicrobial must be lower than that of the susceptible form.

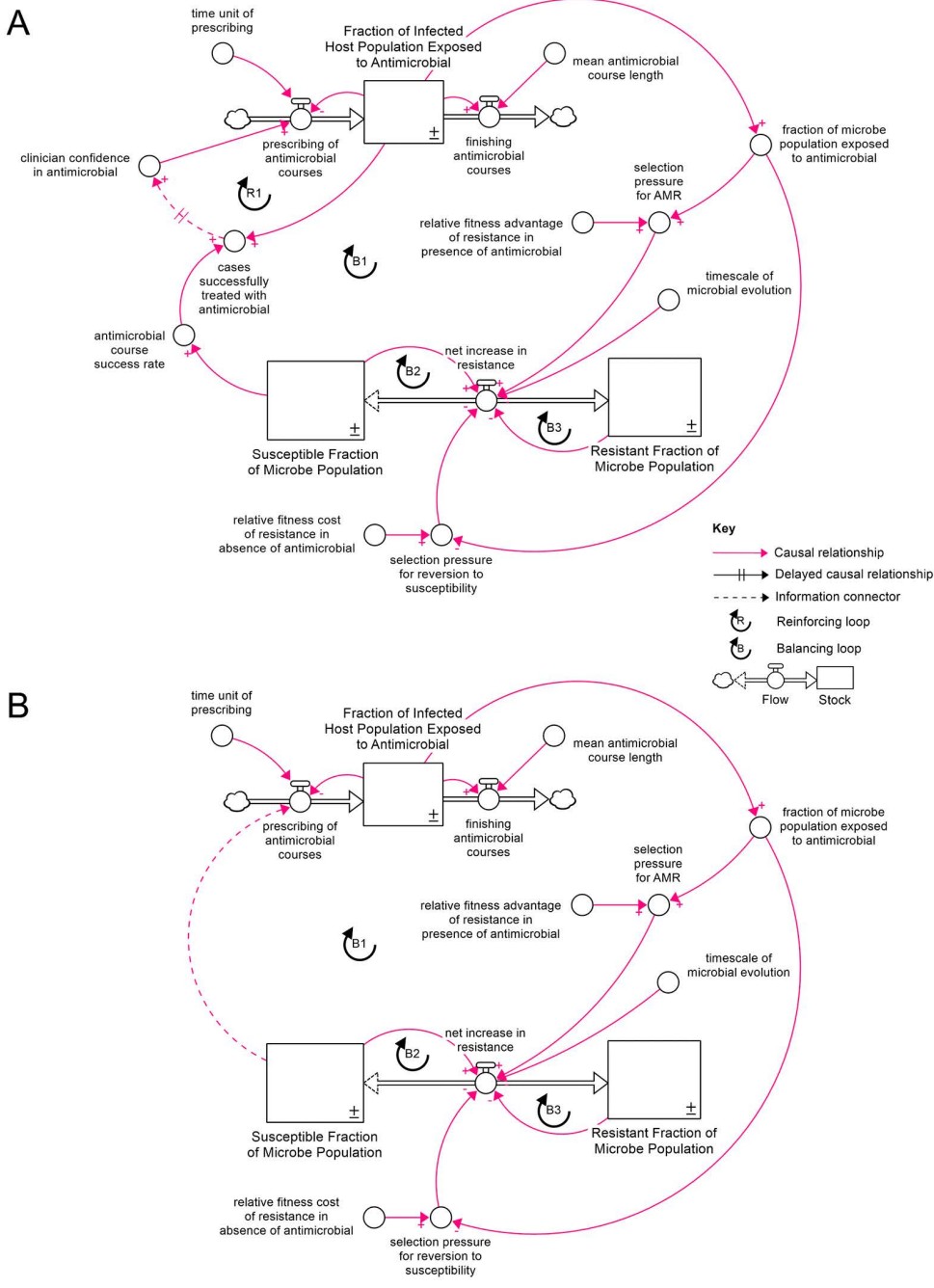

**Fig 3. System Dynamics model structure representing AMR as a Limits to Growth archetype.** Structure pictured a) in its basic form, with prescribing determined by anecdotal success with the antimicrobial, and b) with surveillance providing an information connector between susceptibility and AMU.

For example, *marR* mutations and others contributing to fluoroquinolone (FQ) resistance in *Escherichia coli* have been associated with significant fitness costs [15]. In cases in which compensatory fitness mutations have not arisen in the population, this could be considered an example of renewable susceptibility. Another example of renewable susceptibility is that of *Neisseria gonorrhoeae* to the third-generation cephalosporin cefixime. Cefixime susceptibility in *N. gonorrhoeae*

isolates from one UK subpopulation increased from its nadir of 67% in 2010 to 99% four years later following removal of this drug as a recommended first-line treatment for uncomplicated gonorrhoea [16].

In general, transmitted via mobile genetic elements such as plasmids tend to carry a lower fitness cost than chromosomal AMR mutations [17]. For example, studies indicate that some extended-spectrum beta-lactamase (ESBL) plasmids in *E. coli* are associated with negligible, if any, fitness costs, and can persist without exposure to antimicrobials [18]. As a result, in *E. coli* populations with these ESBL plasmids, susceptibility to beta-lactam antibiotics could be viewed as non-renewable. Another apparent example of non-renewable susceptibility is that of *E. coli* to trimethoprim: a study in Sweden found no measurable fitness cost of resistance *in vitro* and no population-level decrease in resistance following a two-year restriction on use of the drug [19].

## Sensitivity testing

Multiparameter sensitivity testing was carried out on both structural variants of the model in order to assess the robustness, reliability and behaviour of the model under different conditions and to explore the influences of different parameter values on behaviour. Testing addressed both modelled time parameters for response to AMU (microbial evolution and clinician judgment) and relative fitness advantage and cost of the AMR mutation in the presence and absence of the antimicrobial, respectively. Heterogeneity among microbes in rate of evolution and among AMR mutations in relative fitness [20] suggest that these parameters could vary substantially under field conditions. The evolutionary fitness landscape and the timeframe of clinician judgment may also be modifiable, with potential implications for mitigation of AMR depending on their effects on system behaviour.

In the analysis of response timescale parameters, the timescale of microbial evolution was investigated for values of 0.1–26 weeks, while the timescale of clinician judgment, a parameter present only in the "anecdotal prescribing" model variant, covered a range of 1–26 weeks. The relative fitness advantage of resistance in the presence of the antimicrobial was set to 0.95, and the cost of resistance to 0.05 in its absence. In the analysis exploring relative fitness, the relative fitness parameters were explored from $1 \times 10^{-14}$ to 1 in both the presence and absence of the antimicrobial, ranging from essentially no advantage or cost conferred by resistance, respectively, to complete dominance or extinction in a single time increment. Depending on the microbe and resistance mutation, there could be no cost associated with resistance, or even a fitness advantage conferred by the resistance mutation in the absence of the antimicrobial [17,21] although the model does not cover this latter possibility, instead focusing on anthropogenic depletion of susceptibility.

These parameter ranges are nevertheless wider than would likely be biologically plausible. Resistance may be expected to confer a sizeable advantage in the presence of the antimicrobial, and empirical studies suggest that while fitness costs of resistance in the absence of antimicrobials vary, they are rarely large enough to cause rapid near-extinction of resistance. For example, a meta-analysis found that antibiotic resistance conferred by plasmids and chromosomal mutations resulted in, on average, 9% and 21% reductions in maximum growth rate respectively [17]. For an antimicrobial to be clinically useful, a substantial reduction in the susceptible microbial population would have to be achieved within a typical antimicrobial course length. Similarly, the timescale of microbial evolution, the denominator of the rate of movement between compartments, is likely to be markedly shorter than the highest value for this parameter used in the sensitivity analysis. Accordingly, a subset of the sensitivity runs with restricted parameter ranges, corresponding to increased biological plausibility, are presented in an additional plot (S1 Fig).

The time horizon of clinician judgment was set to 6 weeks and the timescale of microbial evolution to 5 weeks. In all analyses, the initial susceptible fraction was set to 1 (a naïve and completely susceptible population), time unit of prescribing to 1 week and mean antimicrobial course length to 2 weeks. The initial microbial population was set to be fully susceptible for simplicity, although in reality, a baseline level of resistance may be expected prior to introduction of a new antimicrobial [3].

 

Uniform distributions of parameter values were explored over runs using Sobol sequencing [22]. However, for the analysis of timescale of microbial evolution under surveillance-informed prescribing, the initial 50 runs indicated that the model behaved differently for parameter values of less than 2 weeks. An additional 25 runs were carried out with the timescale of microbial evolution varying from 0.1–2 weeks, with all other parameters the same, in order to clarify model behaviour over this range. Runs of 2080 weeks (40 years) were carried out in order to assess time for equilibration in the "anecdotal" variant of the model, while 520 weeks (10 years) was more than sufficient for equilibration in the surveillance-informed version.

Model behaviour was examined for each sensitivity analysis, in particular the time taken to reach equilibrium, oscillations in susceptibility prior to stabilisation, and final susceptible fraction of the microbe population.

## Results

### Model structure

The initial iteration of this model closely followed the Limits to Growth archetype, with one reinforcing loop and one balancing loop (Figs 1 and 2a). In this model, use of an antimicrobial to treat a susceptible infection results in successful treatment of cases, increasing confidence in the drug (reinforcing loop R1). This encourages further use of the drug, but AMU creates selection pressure for resistance – essentially "harvesting" the natural resource of antimicrobial susceptibility. With depletion of susceptibility, the number of cases successfully treated with the drug decreases, reducing confidence in the drug and thus driving use of antimicrobials other than the one considered in the model (balancing loop B1). The extent and rate of reversion of the microbial population to susceptibility as a result of curtailed AMU depends on the fitness cost of the resistance mutation in question. There are two sources of delay in this model structure: the time taken for the susceptible fraction of the microbe population to change in response to changes in selection pressure, and the time horizon over which confidence in the antimicrobial is determined.

In a departure from the archetypal structure, an additional information connector may be added to represent surveillance of AMR, bypassing the reinforcing loop R1 (Fig 2b) so that use of the drug in question is determined by susceptibility rather than the anecdotal confidence in the drug driven by the number of cases successfully treated recently. This eliminates the delay created by anecdotal judgment of antimicrobial efficacy.

From these two CLDs, a pair of quantitative SD models were developed, with and without surveillance of AMR bypassing the reinforcing loop R1 (Fig 3). Microbial reversion to the susceptible wild-type was represented with changes in resistant and susceptible fractions of the population dependent on their current values (see Equations 1 and 2) to reflect the fact that in a population with a resistance mutation present at a very low gene frequency, only a small proportion of the population is capable of reverting to wild-type. The effect of this is to create two additional balancing loops, B2 and B3, with the potential to stabilise the system and reduce the presence of oscillations in antimicrobial susceptibility in comparison to many LTG systems.

### Model equations and variables

Equations 1–8. Model equations. Note that Equations 5 and 8 are present in the anecdotal prescribing version of the model only, while in the surveillance-based variant, 5 is replaced by 6. Variables are presented alongside the biological and behavioural phenomena they represent in Table 1.

$$\frac{dS}{dt} = \frac{(P_S \times R) - (P_R \times S)}{M} \tag{1}$$

**Table 1. Model variables and the biological or behavioural phenomena they represent.**

| Variable | Biological or behavioural process/parameter |
|---|---|
| $S$, susceptible fraction of microbe population (0–1; dimensionless) | Fraction of the total microbial population (independent of hosts) that is susceptible to the modelled antimicrobial. In this model, this fraction is assumed to be equal to the antimicrobial course success rate. |
| $R$, resistant fraction of microbe population (0–1; dimensionless) | Inverse of $S$. |
| $F_P$, relative fitness advantage of resistance in presence of antimicrobial (0–1; dimensionless) | Fitness advantage of the resistant genotype when exposed to the antimicrobial, quantified as the fraction of the susceptible population that will be replaced (through a combination of *de novo* mutation, horizontal gene transfer and competition from resistant microbes) by the resistant form at the following timepoint. |
| $F_A$, relative fitness cost of resistance in absence of antimicrobial (0–1; dimensionless) | Fitness cost of the resistant genotype in the absence of the antimicrobial, quantified as the fraction of the resistant population that will be replaced (through a combination of loss of function mutations, plasmid clearance, and competition from susceptible microbes) by the susceptible wild-type at the following timepoint. |
| $E$, fraction of infected host population exposed to antimicrobial (0–1; dimensionless) | Fraction of infected host population exposed to the antimicrobial; in this model this is assumed to be equal to the fraction of microbe population exposed to antimicrobial, assuming equal microbial density among infected individuals. |
| $P_S$, selection pressure for reversion to susceptibility (0–1; dimensionless) | Selection pressure for reversion to susceptibility affecting the overall microbial population as a whole, quantified as the relative fitness cost of the resistant form in the absence of the antimicrobial multiplied by the fraction of the population that is not currently exposed. |
| $P_R$, selection pressure for development of resistance (0–1; dimensionless) | Selection pressure for development of resistance affecting the overall microbial population as a whole, quantified as the relative fitness advantage of the resistant form in the presence of the antimicrobial multiplied by the fraction of the population that is currently exposed. |
| $J$, clinician/farmer/veterinarian [judgement of] confidence in antimicrobial (0–1; dimensionless) | Probability that a potential user/prescriber of the modelled antimicrobial will select this drug for treatment. Anecdotal confidence is modelled here as a smooth stock variable, changes in which are determined by the number of cases successfully treated with the drug over the time horizon of clinician judgment. |
| $C$, cases successfully treated with antimicrobial (0–1; dimensionless) | The fraction of total cases of the disease that are successfully treated with the modelled antimicrobials. |
| $H_J$, time horizon of clinician judgment (weeks) | The time period over which the potential user of the modelled antimicrobial considers their experience of success with the drug. |
| $H_E$, time unit of prescribing (weeks) | Coefficient for conversion of prescribing timescale into weeks (for example, for converting annually reported AMU data into number of prescriptions per week). |
| $M$, timescale of microbial evolution (weeks) | The denominator of the rate of change in microbial susceptibility in response to current fraction of microbial population exposed to the antimicrobial. |
| $L$, mean antimicrobial course length (weeks) | The mean duration of exposure to antimicrobials for any given antimicrobial course. |

$$\frac{dR}{dt} = \frac{(P_R \times S) - (P_S \times R)}{M}$$

(2)

$$P_S = (1 - E) \times F_A$$

(3)

$$P_R = E \times F_P$$

(4)

Anecdotal prescribing model only:

$$\frac{dE}{dt} = \frac{J \times (1 - E)}{H_E} - \frac{E}{L}$$

(5)

Surveillance-informed prescribing model only:

$$\frac{dE}{dt} = \frac{S \times (1 - E)}{H_E} - \frac{E}{L}$$

(6)

$$C = E \times S$$

(7)

Anecdotal prescribing model only:

$$\frac{dJ}{dt} = \frac{(C - J)}{H_J}$$

(8)

**3.2.1. Assumptions.** The model emphasises the LTG framing of AMU and AMR for conceptual exploration rather than a detailed representation of microbial evolution, and thus contains a number of simplifications. Susceptibility, $S$, is assumed to be equal to the probability of treatment success, assuming perfect efficacy and no treatment failure unrelated to AMR. A single bug-drug-mutation-context combination is considered and the model assumes that a single mutation is responsible for AMR for this combination of bug, drug, and context. The microbial population is modelled independently of the hosts, with no reservoirs, and all infected hosts are assumed to be treated, whether with the modelled antimicrobial or another. This model also assumes that the host population is homogenous and well-mixed. Selection for resistance is modelled as depending solely on exposure to the antimicrobial, with no effect of microbial population size, drug dose, or course length. The simplification represented by these assumptions facilitates exploration of the dynamics of AMU and AMR as part of a LTG system.

There is assumed to be no intermediate resistance because the model considers only a single mutation; the relative fitness advantage of resistance in the presence of the antimicrobial can be set to a lower value to represent partial resistance. In order to focus on the influence of anthropogenic antimicrobial exposure, the resistant form is assumed to have, at maximum, equal fitness to the wild type, rather than having an overall advantage. There is also assumed to be

no compensatory evolution to modify the fitness parameters of the resistant form during the time course of the model. Compensatory evolution and stepwise acquisition of resistance do occur in some examples of AMR but are by no means ubiquitous [20]. These phenomena were excluded from this conceptual model, but could be incorporated for applications of the model to specific pathogens and mutations.

The model represents microbial population dynamics with partial symmetry between the S and R compartments, with differing fractional replacement rates for the two forms of the microbe, but both represented with replacement rates proportional to relative fitness parameters.

Anecdotal confidence in the antimicrobial is modelled as a smooth stock variable to represent the probability that any one prescriber will select the antimicrobial in question based on anecdotal success, without use of diagnostics for microbial sensitivity testing. Whilst the hypothetical clinician may have access to microbial sensitivity testing, antimicrobials are frequently prescribed while awaiting culture and sensitivity results. This anecdotal confidence variable occurs only in the version of the model without surveillance, assuming a clinician (or other user of the antimicrobial) with some degree of clinical freedom and no access to surveillance data. In the surveillance-informed version of the model, prescribing probability was kept equal to the current susceptibility frequency to facilitate comparison with the anecdotal prescribing model. Other behavioural drivers of prescribing behaviour, such as cost, are omitted in order to focus on the feedback dynamics.

### Sensitivity analysis

Sensitivity runs for both variants of the model, testing model stability and investigating changing timescale and microbial relative fitness parameters, showed impacts on oscillatory behaviour, time to stabilise, and final susceptible fraction (Fig 4).

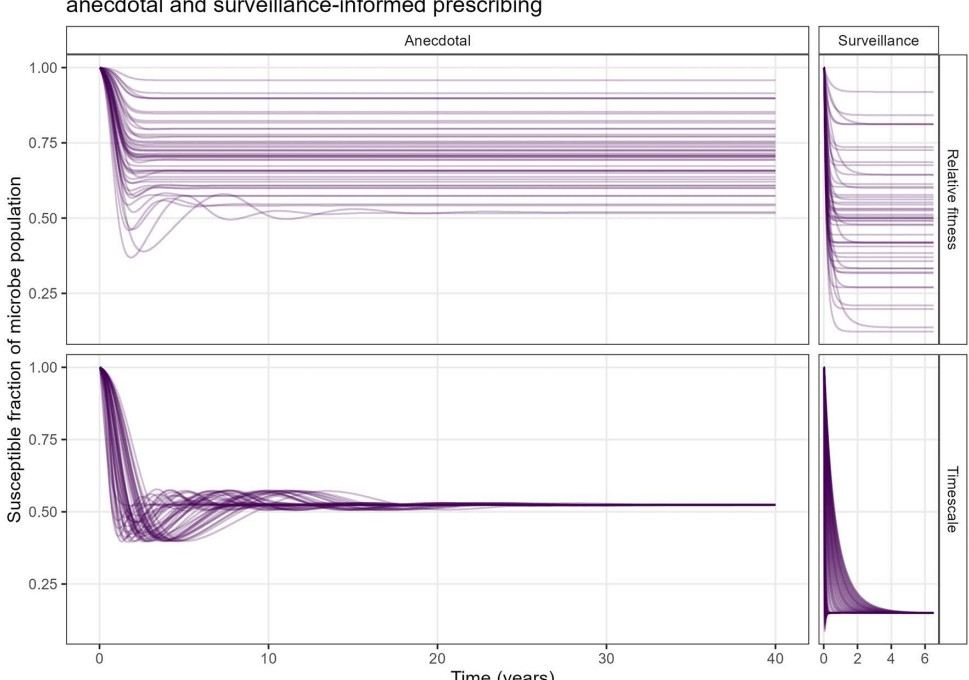

**Fig 4. Sensitivity analysis runs.** Runs varying timescale parameters (time horizon of clinician judgment and timescale of microbial evolution) and relative fitness of the resistant form in presence and absence of the antimicrobial, for both variants of the model structure.

A subset of runs across a biologically plausible parameter subrange is presented in S1 Fig, with $M$ limited to a maximum of 6 weeks, $F_A$ to a maximum of 0.5 and $F_P$ to a minimum of 0.5. Sensitivity analysis results are summarised in Table 2.

Under anecdotal prescribing, both timescale of microbial evolution and time horizon of clinician judgment affected the time taken to stabilise, although the timescale of microbial evolution had a clearer effect (S2 Fig). Both contributed similarly to oscillatory behaviour, with the highest amplitude of oscillations occurring when both timescale parameters were equal. Frequency of oscillations was highest when time parameters were shorter, particularly the timescale of microbial evolution.

Under surveillance-informed prescribing, the parameter reflecting clinician judgment was removed, leaving only microbial evolution (Fig 5). Oscillations occurred only with a timescale of microbial evolution shorter than 1.65 weeks, with stabilisation time increasing linearly with parameter values above this. As the timescale of microbial evolution decreased below this threshold, the number and maximum amplitude of oscillations increased.

Model structure influenced stabilisation time, presence of oscillations and final susceptible fraction. Under anecdotal prescribing, the time to stabilise was observed to range from 105.5 to 2061.75 weeks (approximately 2–40 years). Under

**Table 2. Summary of sensitivity analysis results.**

| Parameter(s) or structural change of interest | Range covered in sensitivity analysis | Effects on model behaviour |
|---|---|---|
| a) time horizon of clinician judgment and b) timescale of microbial evolution | 1-26 weeks for time horizon of clinician judgment (anecdotal prescribing structure only)<br>0.1-26 weeks for timescale of microbial evolution | Anecdotal prescribing (S2 Fig):<br>• Both parameters affected the time taken to stabilise, especially timescale of microbial evolution<br>• Both contributed similarly to oscillatory behaviour<br>• Oscillation frequency decreased with increasing time parameter values<br>Surveillance-informed prescribing (Fig 5):<br>• Oscillations seen only with timescale of microbial evolution under 1.65 weeks<br>• Number and amplitude of oscillations decreased as parameter values increased toward this threshold |
| Relative fitness of resistant form in a) presence and b) absence of antimicrobial | $1 \times 10^{-14}$–1 for relative fitness of resistant form under both conditions | Anecdotal prescribing (S3 Fig):<br>• Oscillations occurred with relative fitness cost of resistance in absence of antimicrobial under approximately 0.52<br>• Stabilisation time determined mainly by relative fitness cost of resistance in absence of antimicrobial<br>Surveillance-informed prescribing (S4 Fig):<br>• No oscillations for any relative fitness combinations<br>• Similar degree of influence of both relative fitness parameters on stabilisation time; lower relative fitness associated with longer stabilisation times for both parameters<br>Both model variants:<br>• Similar degree of influence of both relative fitness parameters on final susceptible fraction at equilibrium<br>• Effects of each parameter on final susceptible fraction were strongest at lower parameter values |
| Basis of prescribing (model structure) | Anecdotal (based on recent success of antimicrobial treatments) versus surveillance-informed (based on current susceptible fraction) | • With anecdotal prescribing, presence and characteristics of oscillations were determined by timescale parameters and relative fitness in the absence of the antimicrobial, while under surveillance-based prescribing, oscillations only occurred at shortest timescales of microbial evolution<br>• Surveillance-informed prescribing much reduced stabilisation time, compared to anecdotal prescribing<br>• Final susceptible fraction of microbe population was generally lower with surveillance-informed prescribing |

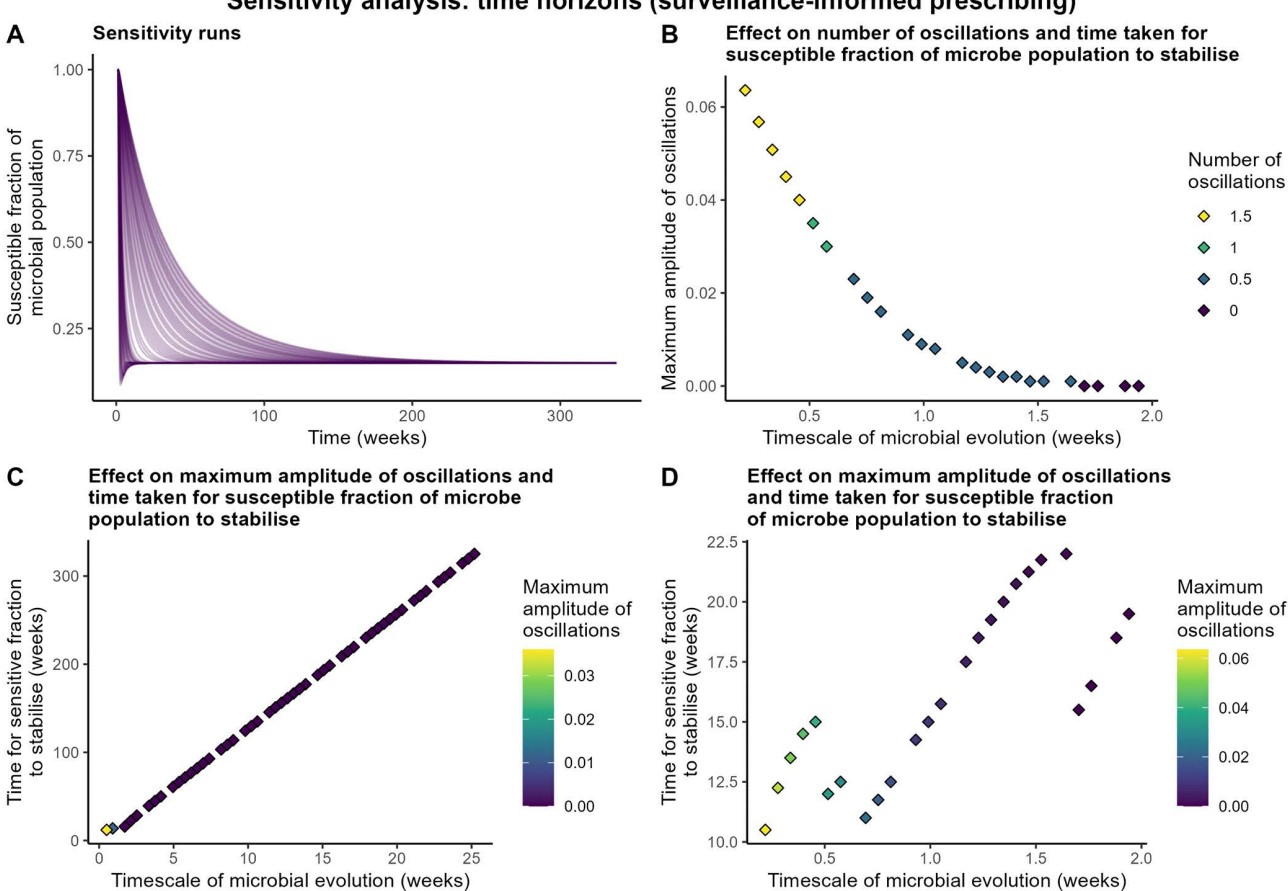

**Fig 5. Sensitivity analysis results.** Results varying timescale of microbial evolution for the surveillance-based prescribing variant of the model: a) sensitivity runs with microbial evolution timescale ranging 0.1-26 weeks, b) effect of parameter values ranging 0.1-2 weeks on number and amplitude of oscillations prior to stabilisation, and effect of parameter value on maximum amplitude of oscillations and time for susceptible fraction to stabilise with parameter ranges of c) 0.1-26 weeks and d) 0.1-2 weeks.

surveillance-informed prescribing, stabilisation time was much reduced, ranging from 10.5 weeks to 325.25 weeks (<1 year to approximately 6 years), and the final susceptible fraction of the microbe population was typically lower. With anecdotal prescribing, presence, number, frequency and amplitude of oscillations varied with timescale parameters and relative fitness cost of the resistant form in the absence of the antimicrobial, while in the surveillance-informed prescribing variant, oscillations were largely eliminated, only occurring at the lowest values of timescale of microbial evolution.

## Discussion

Considering antimicrobial susceptibility as a natural resource, and specifically the application of LTG modelling, offers a useful framing of the global challenge of AMR. AMR, a natural phenomenon occurring as part of the evolutionary arms race between microbes, existed long before humans started to use antimicrobials [3]. Representing susceptibility as a common property resource to be conserved and sustainably managed has great potential for engagement with the public and policymakers and avoids widely-used war metaphors for AMR that risk miscommunicating the problem and its potential solutions [23]. The use of SD modelling is of additional value in that it incorporates time lags and allows comparison of model behaviour over time under different conditions.

A particular advantage of the LTG model is that it explicitly incorporates both human behaviour and evolutionary dynamics, allowing an exploration of the role of AMR surveillance and facilitating hypothesis generation regarding phenomenon of plateauing AMR observed in clinical surveillance [7]. A number of evolutionary explanations have been proposed [6], but to our knowledge, this is the first model to suggest a hypothetical behavioural explanation where those using the antimicrobial in question lose confidence in its efficacy and therefore decrease their use of the drug.

Behavioural mechanisms represent one possible contributor among many, therefore further research is required to distinguish between causes. Multiple evolutionary explanations may contribute, such as host population structure, including variable rates of antimicrobial use and differential transmission patterns between hosts [24]. Other potential evolutionary explanations include the proximity of resistance genes to the loci of genes subject to negative frequency-dependent (balancing) selection and the transient resistance-favouring niche represented by treated members of the host population [6]. None of the above explanations would be expected to produce oscillations prior to stabilisation, providing a possible way to distinguish causes from empirical observations. In the sensitivity analysis, oscillations occurred in scenarios with a relatively low fitness cost of resistance. Meta-analysis of empirical studies suggest that the fitness cost of resistance is usually small, suggesting that oscillations might be expected to occur in the field if AMU is determined by prescriber confidence [17]. The wavelength of oscillations seen in the sensitivity analysis for anecdotally-informed prescribing ranged from 138 to 813 weeks (138–454 weeks in the restricted parameter range plotted in S1 Fig), such that population-level sampling carried out annually [25] over a period of 3–16 years (depending on parameters) in these contexts would be expected to detect oscillations if they were present. However, when prescribing was surveillance-informed and microbial evolution was very rapid, oscillations occurred with a wavelength of as little as 7 weeks, meaning that they would be missed by annual sampling for AMR surveillance and could require sampling every 3–4 weeks for detection.

Comparison with historical examples is challenging particularly for the anecdotal prescribing variant of the model, which describes settings without surveillance: that is, ones lacking widely available population-level data. Where data are available, surveillance is frequently only available for limited timespans, shorter than those covered by most of the oscillations in the model's sensitivity analysis and beginning well after the initial introduction of the antimicrobial to a naïve population.

Comparison with empirical values for fitness is also challenging, as the conceptualisation of relative fitness in this model uses net displacement rates is therefore not directly comparable with the relative fitness parameter typically reported for the fitness cost of a resistant strain *in vitro*. These reported *in vitro* parameters are informed by competition experiments or comparisons of proxy fitness measures such as the Malthusian parameter [17]. This model's representation of fitness advantages and costs uses net fractional displacement rates, aggregating *de novo* mutations, horizontal gene transfer (HGT), and vertical gene transmission, as well as differences in growth rate and infection clearance rate. In order to parameterise the model directly from empirical fitness data, the microbial evolution portion of the model could be reconfigured to include one Malthusian relative fitness parameter for the resistant form in the absence of the antimicrobial and another in the concentrations of antimicrobial typically achieved during antimicrobial therapy. Additional parameters could include the microbial mutation rate and rate of HGT.

Although the model represents the "flows" between S and R as similar in both directions, the processes of gain and loss of AMR genes and population-level dynamics may not be symmetrical. Ecological analysis has observed asymmetry in AMU-AMR response times. Delays may occur between curtailing population-level AMU and reduction in AMR prevalence, while a rise in AMR may occur immediately after initiating AMU [26]. Partial model calibration using paired AMU and AMR time series would facilitate testing of the evolutionary portion of this model to assess the impact of the simplifications on the model's ability to approximate the AMU-AMR relationship.

The simplicity of this model has both advantages and disadvantages: it can potentially be adapted to different microbes, antimicrobials, host species and One Health settings. However, it necessarily simplifies the evolutionary dynamics of AMR, focusing on the population-level effects of gene frequency of a single AMR mutation rather than considering different dynamics of *de novo* mutations *versus* vertical and HGT, and focuses on a single resistance

mutation. The model also assumes that all infected hosts are symptomatic and receive treatment with this antimicrobial or another, and that all microbes are in the host population with no reservoirs of infection. Selection pressure for resistance is modelled as depending solely on the binary of exposure/non-exposure to the antimicrobial, with no effect of co-selection for AMR or inappropriate course lengths or doses, or indeed total microbial population size. In the surveillance-informed model variant, the probability of prescribing the antimicrobial is modelled as equal to the prevalence of susceptibility, but for more realism in this model variant, policy-driven threshold-based switching of antimicrobial choice could be considered as an alternative. A real-life example of this is the UK policy, based on WHO guidelines, of switching away from cefixime to treat *N. gonorrhoeae* infections once population-level resistance prevalence reached 5%, allowing susceptibility to "renew" [16]. The assumption of clinical freedom regarding antimicrobial drug choice is more applicable to some contexts than others: anecdotal prescribing behaviours modelled here are likely to have much less effect on AMU and AMR dynamics when choice of antimicrobial is determined largely by national or organisational prescribing policy, as in the above example.

Relatedly, neither variant of the model includes behavioural factors beyond confidence in antimicrobial efficacy. These could include prescribing incentives, drug availability and cost, and in human clinical settings, patient demand. In real-world scenarios, these factors might be expected to reduce use of certain antimicrobials even when susceptibility is high, as well as influencing choices between antimicrobials with similar "stocks" of susceptibility. In some circumstances, cost or drug availability might preclude the use of any antimicrobial treatment, not only the one being modelled; this could influence infectious disease burden and thus dynamics of resistance within the microbial population. The applicability and relative importance of these biological and behavioural phenomena, as well as parameter values, are likely to vary substantially between applications of the model, so we ensured that our sensitivity analysis explored a wide parameter space for both model variants.

Future iterations of this model could address some of these assumptions. For policy translation it would be necessary, for example, to consider wider behavioural drivers of prescribing behaviour. One simple way to do this would be by including an additional parameter summarising overall drug "appeal" in terms of cost and prescribing incentives. Surveillance-informed threshold-based switching of prescribing guidelines could also be introduced, with parameters to adjust prescribers' adherence to the guidelines *versus* anecdotal confidence and other behavioural drivers.

In the sensitivity analysis, oscillations in susceptibility were generally restricted to the version of the model with anecdotally-informed AMU. In this variant of the model, the relative fitness cost of resistance in the absence of antimicrobials showed a much greater effect on oscillatory behaviour than the degree of advantage conferred by the resistance mutation on exposure to the antimicrobial. Notably, oscillations occurred when the displacement rate due to fitness cost was less than approximately 0.5.

Timescale parameters had comparable levels of impact on oscillatory behaviour, with oscillation frequency decreasing with increasing time parameter values. When oscillations in susceptibility did occur, the system reached a steady state eventually, with relative fitness advantage and cost of resistance in the presence and absence of the antimicrobial, respectively, showing similar degrees of effect on the final susceptible fraction and both timescale parameters, particularly microbial evolution, influencing the time taken to stabilise. It is possible that the deterministic nature of this model is responsible for the dampening of oscillations; in other compartmental models, stochastic perturbations have been shown to counteract this [27]. Deterministic modelling may also underestimate the stochastic processes that are important for AMR dynamics at very low gene frequencies or in small population sizes. The deterministic conceptualisation of relative fitness used here, quantified as overall rate of change of susceptible and resistant fractions, assumes that extinction of either microbial variant will always be counteracted by re-emergence. The model's representation of microbial ecology could be enhanced with incorporation of both stochasticity and differential microbial growth rates as modelled by Homer *et al.* [13]. Furthermore, differences in population dynamics could be considered between different types of microbes and between the effects of antimicrobials that directly kill microbes *versus* those that inhibit reproduction. A partial model

calibration could be used to test the microbial evolution component of the model using population level data on AMU and AMR [13].

The surveillance-informed AMU variant of the model largely eliminated oscillations, which only occurred in this variant under conditions of rapid microbial evolution. In this version of the model structure, antimicrobial susceptibility reached a steady state much sooner, and typically with a lower final susceptible fraction. This could be viewed as a net positive outcome: the fraction of infections susceptible to the antimicrobial is lower, but the antimicrobial is being used as much as possible on the susceptible cases, while leaving a "reserve" of susceptibility in the microbial population. In order for the correct patients (those with infections susceptible to the antimicrobial) to receive the drug in question, individual sensitivity testing would be optimal rather than population-level surveillance. Nevertheless, this illustrates the utility of population-level AMR surveillance and implies a greater-still value of diagnostics for individual patients.

## Conclusion

The model presented in this conceptual study applies a well-characterised modelling approach in a novel context, using a system archetype commonly explored in SD to model both behavioural and ecological aspects of AMR in both human and veterinary contexts.

The model variants, one following the classic LTG structure and one eliminating the human decision-making component, allows for analysis of two scenarios over a wide parameter space. Multiparameter sensitivity analysis of both variants revealed a dramatic decrease in oscillations and time taken to stabilise when surveillance, rather than recent experience of treatment success, determined AMU. Surveillance-informed AMU was also associated with a lower final susceptible fraction of the microbe population, although the final fraction was also influenced by the relative fitness advantage and cost of resistance in the presence and absence of the antimicrobial in question, respectively. Under conditions of anecdotally-informed prescribing, the fitness cost of the mutation in the absence of the antimicrobial had a greater effect on oscillatory behaviour than the evolutionary advantage conferred by resistance in the presence of the antimicrobial, with this oscillations and stabilisation time also influenced by the timescales of microbial evolution and of human judgment of confidence in the antimicrobial.

The socio-ecological model structure facilitates generation of novel hypotheses, including the possibility that AMR could plateau as a result of human behaviour as opposed to solely evolutionary forces [6], and a revival of an underappreciated framing of antimicrobial susceptibility as a renewable natural resource to be conserved, recognising AMR as a natural evolutionary phenomenon. This framing is valuable not only as an avenue for alternative modelling approaches, but also as an alternative, more collaborative, framing for public and policy engagement to promote sustainable management of this common pool resource.

## Supporting information

**S1 Table. Settings used for sensitivity analysis.**
(DOCX)

**S1 Fig. Sensitivity runs: biologically plausible subset of parameter values.**
(TIF)

**S2 Fig. Sensitivity analysis findings: time horizon parameters (anecdotal prescribing).**
(TIF)

**S3 Fig. Sensitivity analysis findings: relative fitness parameters (anecdotal prescribing).**
(TIF)

**S4 Fig. Sensitivity analysis findings: relative fitness parameters (surveillance-informed prescribing).**
(TIF)

**S1 File. Supporting information README document.**
(DOCX)

**S2 File. Sensitivity run outputs.**
(XLSX)

**S3 File. ZIP file archive containing model files.**
(ZIP)

## Acknowledgments

The authors are grateful to both reviewers for their very constructive recommendations, which we feel have substantially improved the paper. CRW would also like to thank Adam Groves for his coding assistance.

## Author contributions

**Conceptualization:** Carys J. Redman-White.

**Data curation:** Carys J. Redman-White.

**Formal analysis:** Carys J. Redman-White.

**Investigation:** Carys J. Redman-White, Birgit Kopainsky.

**Methodology:** Carys J. Redman-White, Birgit Kopainsky.

**Supervision:** Birgit Kopainsky, Adrian Muwonge, Andrew R. Peters, Dominic Moran.

**Visualization:** Carys J. Redman-White.

**Writing – original draft:** Carys J. Redman-White, Dominic Moran.

**Writing – review & editing:** Carys J. Redman-White, Birgit Kopainsky, Adrian Muwonge, Andrew R. Peters, Dominic Moran.

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
