## [Decision Letter · Decision Letter 0]

8 Feb 2026

Dear Dr. Redman-White,

Thank you for submitting your manuscript to PLOS ONE. After careful consideration, we feel that it has merit but does not fully meet PLOS ONE’s publication criteria as it currently stands. Therefore, we invite you to submit a revised version of the manuscript that addresses the points raised during the review process.

A letter that responds to each point raised by the academic editor and reviewer(s). You should upload this letter as a separate file labeled ‘Response to Reviewers’.A marked-up copy of your manuscript that highlights changes made to the original version. You should upload this as a separate file labeled ‘Revised Manuscript with Track Changes’.An unmarked version of your revised paper without tracked changes. You should upload this as a separate file labeled ‘Manuscript’.

If applicable, we recommend that you deposit your laboratory protocols in protocols.io to enhance the reproducibility of your results. Protocols.io assigns your protocol its own identifier (DOI) so that it can be cited independently in the future. For instructions see: https:://journals.plos.org/plosone/s/submission-guidelines#loc-laboratory-protocols. Additionally, PLOS ONE offers an option for publishing peer-reviewed Lab Protocol articles, which describe protocols hosted on protocols.io. Read more information on sharing protocols at . Additionally, PLOS ONE offers an option for publishing peer-reviewed Lab Protocol articles, which describe protocols hosted on protocols.io. Read more information on sharing protocols at . Additionally, PLOS ONE offers an option for publishing peer-reviewed Lab Protocol articles, which describe protocols hosted on protocols.io. Read more information on sharing protocols at . Additionally, PLOS ONE offers an option for publishing peer-reviewed Lab Protocol articles, which describe protocols hosted on protocols.io. Read more information on sharing protocols at https:://plos.org/protocols?utm_medium=editorial-email&utm_source=authorletters&utm_campaign=protocols....

We look forward to receiving your revised manuscript.

Kind regards,

Md. Tanvir Rahman, DVM, MSc (Canada), PhD (UK), FBAS

Academic Editor

PLOS One

Journal Requirements:

https:://journals.plos.org/plosone/s/file?id=wjVg/PLOSOne_formatting_sample_main_body.pdf and

https:://journals.plos.org/plosone/s/file?id=ba62/PLOSOne_formatting_sample_title_authors_affiliations.pdf

“CRW acknowledges support for a studentship part-funded by Zoetis and the UKRI Biotechnology and Biological Sciences Research Council (BBSRC) under grant number BB/T00875X/1. DM also acknowledges support under UKRI awards UKRI (BB/Z515644/1, BB/T004436/1 and BB/T004452/1, the latter of which is [co] funded by the UK Department of Health and Social Care as part of the Global AMR Innovation Fund (GAMRIF). This is a UK aid programme that supports early-stage innovative research in underfunded areas of antimicrobial resistance (AMR) research and development for the benefit of those in low- and middle-income countries (LMICs), who bear the greatest burden of AMR. The views expressed in this publication are those of the author(s) and not necessarily those of the UK Department of Health and Social Care. AM is funded by the University of Edinburgh Chancellor's Fellowship and BBSRC core funding for the Roslin Institute.”

“CRW acknowledges support for a studentship part-funded by Zoetis and the UKRI Biotechnology and Biological Sciences Research Council (BBSRC) under grant number BB/T00875X/1. DM also acknowledges support under UKRI awards UKRI (BB/Z515644/1, BB/T004436/1 and BB/T004452/1, the latter of which is [co] funded by the UK Department of Health and Social Care as part of the Global AMR Innovation Fund (GAMRIF). This is a UK aid programme that supports early-stage innovative research in underfunded areas of antimicrobial resistance (AMR) research and development for the benefit of those in low and middle-income countries (LMICs), who bear the greatest burden of AMR. The views expressed in this publication are those of the author(s) and not necessarily those of the UK Department of Health and Social Care. AM is funded by the University of Edinburgh Chancellor's Fellowship and BBSRC core funding for the Roslin Institute. For the purpose of open access, the authors have applied a Creative Commons Attribution (CC BY) licence to any Author Accepted Manuscript version arising from this submission.”

“CRW acknowledges support for a studentship part-funded by Zoetis and the UKRI Biotechnology and Biological Sciences Research Council (BBSRC) under grant number BB/T00875X/1. DM also acknowledges support under UKRI awards UKRI (BB/Z515644/1, BB/T004436/1 and BB/T004452/1, the latter of which is [co] funded by the UK Department of Health and Social Care as part of the Global AMR Innovation Fund (GAMRIF). This is a UK aid programme that supports early-stage innovative research in underfunded areas of antimicrobial resistance (AMR) research and development for the benefit of those in low- and middle-income countries (LMICs), who bear the greatest burden of AMR. The views expressed in this publication are those of the author(s) and not necessarily those of the UK Department of Health and Social Care. AM is funded by the University of Edinburgh Chancellor's Fellowship and BBSRC core funding for the Roslin Institute.”

5. We are unable to open your Supporting Information file “Model files.zip”. Please kindly revise as necessary and re-upload.

Additional Editor Comments:

Please address the comments.

Reviewer's Responses to Questions

**Comments to the Author**

1. Is the manuscript technically sound, and do the data support the conclusions?

Reviewer #1: Yes

Reviewer #2: Yes

2. Has the statistical analysis been performed appropriately and rigorously?

Reviewer #1: Yes

Reviewer #2: Yes

3. Have the authors made all data underlying the findings in their manuscript fully available?

Reviewer #1: Yes

Reviewer #2: Yes

4. Is the manuscript presented in an intelligible fashion and written in standard English?

Reviewer #1: Yes

Reviewer #2: Yes

Reviewer #1: The manuscript frames AMR as a Limits-to-Growth system and develops two complementary causal-loop diagrams (CLDs) and quantitative system-dynamics (SD) models: an anecdotal prescribing variant in which antimicrobial use (AMU) is driven by recent anecdotal success, and a surveillance-informed variant in which measured susceptibility feeds back to prescribing. The authors explore model behaviour across wide parameter ranges (microbial evolution timescales, clinician judgment horizon, and resistant fitness) with Sobol-sequence sampling and long runs (up to 40 years) to compare equilibria, oscillations, and time-to-stabilization between structures.

1. Casting AMR in an LTG frame is conceptually apt: AMU “harvests” susceptibility as a limiting resource and feedbacks to prescribing behaviour can generate oscillations and long transients. This framing will resonate with policy audiences and systems-modelling readers.

2. The contrast between anecdotal prescribing and surveillance-informed prescribing captures an important policy question — whether information feedbacks shorten harmful oscillatory dynamics and accelerate system stabilization.

3. The authors explore a wide parameter space (timescales, fitness) and report on qualitative features (oscillations, stabilisation time, equilibrium susceptible fraction), which is appropriate for exploratory SD work.

4. The finding that surveillance can dramatically reduce time to stabilisation is potentially policy-relevant and motivates investment in surveillance infrastructure.

Comments

1. The model treats AMU (antimicrobial use/effort) as exogenous or driven only by recent anecdotal success or by surveillance. The manuscript acknowledges this limitation but does not adequately justify treating AMU as independent of socio-behavioural drivers (costs, access, incentives, patient demand, prescribing norms), which are central determinants of AMR trajectories in real settings. Either (a) explicitly justify why a simplified AMU driver is appropriate for the questions posed (i.e., focusing on feedback form rather than full behavioural realism), or (b) extend the CLD/SD structure to include a simple behavioural module (e.g., prescribing incentives, patient demand, drug availability) and test sensitivity. At minimum, discuss how exclusion of these factors affects interpretation and policy translation.

2. The SD models appear entirely theoretical — parameter ranges are wide and not calibrated to empirical settings. Without calibration, it is hard to assess whether the oscillatory regimes or stabilization times are realistic for real pathogens/contexts. Provide at least one calibration/validation example using empirical data (even qualitative) — e.g., match model behaviour to known historical AMR dynamics for a pathogen/antimicrobial pair, or show how parameter choices would map to a plausible pathogen (e.g., E. coli, S. aureus) with references for mutation rates and fitness costs. If empirical calibration is infeasible, clearly label the study as conceptual and include guidance on how to adapt parameters for applied scenarios.

3. Parameter ranges (e.g., relative fitness from 1×10-14 to 1, microbial evolution timescale 0.1–26 weeks) are very broad; however, biological plausibility for extremes is not discussed. Some extremes (1×10-14) are effectively zero and may cause numerical issues. Justify choice of ranges with citations or sensitivity rationale. Consider restricting ranges to biologically plausible bounds for typical bacterial pathogens, and include a rationale for exploring extreme values (e.g., to test numerical stability). Report sensitivity results both across the full exploratory range and within a biologically plausible subrange.

5. The current deterministic SD approach may understate stochastic effects important when resistant alleles are at very low frequency, or when population sizes are small. The manuscript discusses low initial gene frequency but implements deterministic reversion processes. Discuss limitations of deterministic modelling in representing stochastic emergence/extinction. Consider supplementing with stochastic simulations (Gillespie or stochastic SD) for scenarios where resistant allele frequencies are rare. Alternatively, include heterogeneity in mutation rates/fitness across strains as an extended sensitivity analysis.

This manuscript offers a clear and compelling systems framing of AMR dynamics and provides useful insights into how information feedback (surveillance) can alter system behaviour relative to anecdotal prescribing. The modelling approach and sensitivity exploration are valuable. However, before publication in a high-impact journal, the manuscript needs stronger empirical grounding, clearer model specification, and explicit treatment of behavioural drivers and stochastic effects that shape real-world AMR dynamics. Addressing these points will substantially increase the rigor, credibility, and policy relevance of the work.

Reviewer #2: This manuscript presents a conceptually interesting and potentially valuable application of System Dynamics (SD) to antimicrobial resistance (AMR), framing antimicrobial susceptibility as a partially renewable natural resource and embedding human prescribing behaviour within a Limits-to-Growth (LTG) archetype. The behavioural interpretation of AMR plateaux is original and thought-provoking. However, in its current form, the paper does not yet meet the standards for methodological clarity, validation, and interpretability required for publication, even in a methods-oriented journal such as PLOS ONE. Substantial revision is required. I ivite the authors to clarify the following concerns:

1. The framing of susceptibility as a renewable or partially renewable natural resource is repeatedly highlighted as a key contribution. However, the manuscript does not clearly specify under what biological conditions susceptibility is truly renewable, versus quasi-renewable or effectively non-renewable. Fitness costs are treated as abstract parameters, but no clear biological mapping is provided (e.g., plasmid-borne resistance, compensatory evolution, clonal replacement). I invite the authors to add a short conceptual subsection explicitly defining what qualifies susceptibility as renewable in this model, which real-world AMR scenarios this framing plausibly applies to and which important AMR scenarios it explicitly does not apply to. Likewise, please provide at least two concrete biological examples (e.g., fluoroquinolone resistance in E. coli vs. ESBL plasmids) to anchor the abstraction. Without this, the “resource extraction” framing risks appearing metaphorical rather than analytical.

2. The SD structure assumes a single bug–drug–mutation–context, homogeneous host population, no environmental or between-host reservoirs and AMU determined either by anecdotal success or population-level surveillance. While simplification is acceptable, the rationale for these specific simplifications is not adequately defended. The authors must explicitly justify why anecdotal confidence is modelled as a smooth stock variable rather than, for example: threshold-based switching, policy-driven constraints and diagnostic-driven decisions.

3. The mathematical formulation lacks clarity and biological interpretability. The system of equations (Equations 1–7) is central, yet several issues remain: equation 1–2: The symmetry between S and R is mathematically neat but biologically strong; mutation, reversion, and replacement are conflated; equation 5a/5b: The prescribing function is not intuitive, and the role of HE as a “time unit of prescribing” is unclear; equation 6: “Cases successfully treated” are defined as C=E×SC = E \times SC=E×S, implicitly assuming: perfect treatment efficacy for susceptible infections and zero efficacy for resistant infections Please provide a table mapping each equation term to a biological or behavioural process, explicitly state all implicit assumptions (perfect efficacy, no partial resistance, no treatment failure unrelated to AMR), clarify why S is equated to treatment success probability, and discuss implications and consider moving full equations to Supplementary Information and simplifying the main text, but with improved explanation.

4. The manuscript argues that human behavioural feedback alone could explain observed AMR plateaux, a strong and interesting claim. However, the model is not calibrated to any empirical dataset, no comparison is made with known evolutionary explanations beyond citation and the conclusion risks being interpreted as causal rather than exploratory. I invite the authors to reframe conclusions to emphasise hypothesis generation, not explanation, explicitly state that behavioural mechanisms are one possible contributor among many and add a paragraph outlining what empirical tests would be required to distinguish behavioural from evolutionary plateau mechanisms. This reframing is essential to avoid overinterpretation.

.

Reviewer #1: No

Reviewer #2: No

To ensure your figures meet our technical requirements, please review our figure guidelines: https:://journals.plos.org/plosone/s/figures

You may also use PLOS’s free figure tool, NAAS, to help you prepare publication quality figures: https:://journals.plos.org/plosone/s/figures#loc-tools-for-figure-preparation.

---

## [Author Response · Author response to Decision Letter 1]

24 Mar 2026

Thank you for the opportunity to revise this submission. We are grateful to both reviewers for their very constructive recommendations, which we feel have substantially improved the paper. We have addressed all comments and hope that you will find the revised manuscript acceptable. Please find responses to all comments in the attached document.

---

## [Decision Letter · Decision Letter 1]

26 Mar 2026

A socio-ecological System Dynamics model of antimicrobial use and resistance

PONE-D-25-47612R1

Dear Dr. Redman-White,

We’re pleased to inform you that your manuscript has been judged scientifically suitable for publication and will be formally accepted for publication once it meets all outstanding technical requirements.

Kind regards,

Md. Tanvir Rahman, DVM, MSc (Canada), PhD (UK), FBAS

Academic Editor

PLOS One

Additional Editor Comments (optional):

Thanks for the revision.

Reviewers' comments:

Reviewer's Responses to Questions

**Comments to the Author**

Reviewer #2: All comments have been addressed

2. Is the manuscript technically sound, and do the data support the conclusions?

Reviewer #2: Yes

3. Has the statistical analysis been performed appropriately and rigorously?

Reviewer #2: Yes

4. Have the authors made all data underlying the findings in their manuscript fully available?

Reviewer #2: Yes

5. Is the manuscript presented in an intelligible fashion and written in standard English?

Reviewer #2: Yes

Reviewer #2: All of the raised concerns have been carefully addressed, resulting in a significant improvement comparing with the previous version. Thank you!

.

Reviewer #2: No

---

## [Editor Report · Acceptance letter]

PONE-D-25-47612R1

PLOS One

Dear Dr. Redman-White,

I'm pleased to inform you that your manuscript has been deemed suitable for publication in PLOS One. Congratulations! Your manuscript is now being handed over to our production team.

You will receive an invoice from PLOS for your publication fee after your manuscript has reached the completed accept phase. If you receive an email requesting payment before acceptance or for any other service, this may be a phishing scheme. Learn how to identify phishing emails and protect your accounts at https:://explore.plos.org/phishing.

Kind regards,

on behalf of

Professor Md. Tanvir Rahman

Academic Editor

PLOS One